# Choices We Make in Times of Crisis

Patrick O. Waeber [1,*,†] , Natasha Stoudmann [2] , James D. Langston [3] , Jaboury Ghazoul [1] , Lucienne Wilmé [4] ,
Jeffrey Sayer [3] , Carlos Nobre [5] , John L. Innes [3] , Philip Fernbach [6] , Steven A. Sloman [7]
and Claude A. Garcia [1,8,*,†]

1   Department of Environmental Systems Science, Institute of Terrestrial Ecosystems, ETH-Zürich,
    8092 Zürich, Switzerland; jaboury.ghazoul@env.ethz.ch
2   School of Technology, Environments and Design, University of Tasmania, Hobart 7001, Australia;
    natasha.stoudmann@utas.edu.au
3   Faculty of Forestry, University of British Columbia, Vancouver, BC V6T 1Z4, Canada;
    james.langston@ubc.ca (J.D.L.); jeffrey.sayer@ubc.ca (J.S.); john.innes@ubc.ca (J.L.I.)
4   World Resources Institute Africa, Madagascar Program, BP 3884, Antananarivo 101, Madagascar;
    lucienne.wilme@wri.org
5   Institute of Advanced Studies, University of São Paulo, São Paulo 05508-060, Brazil; cnobre.res@gmail.com
6   Leeds School of Business, University of Colorado, Boulder CO 80309, USA; philip.fernbach@colorado.edu
7   Cognitive, Linguistic & Psychological Sciences, Brown University, Providence, RI 02912, USA;
    Steven_Sloman@brown.edu
8   CIRAD, UPR Forêts et Sociétés, CEDEX 5, 34398 Montpellier, France
*   Correspondence: patrick.waeber@usys.ethz.ch (P.O.W.); claude.garcia@usys.ethz.ch (C.A.G.)
†   These authors contributed equally to this work.

**Abstract:** We present a new framework that allows understanding those we deem irrational in the climate debate. Realizing if the issue is one of information, beliefs, values or means opens the door for more constructive dialogue. Decision-makers diverge in their responses to the urgent need for action on climate and biodiversity. Action gaps are fueled by the apparent inability of decision-makers to respond efficiently to the mounting threats described by scientists—and increasingly recognized by society. Surprisingly, with the growing evidence and the accumulation of firsthand experiences of the impacts of environment crises, the gap is not only a problem of conflicting values or beliefs but also a problem of inefficient strategies. Bridging the gap and tackling the growing polarization within society calls for decision-makers to engage with the full complexity of the issues the world is facing. We propose a framework characterizing five archetypes of decision-makers to help us out of the current impasse by better understanding the behavior of others. Dealing with the complexity of environmental threats requires decision-makers to question their understanding of who wins and who loses, and how others make decisions. This requires that decision-makers acknowledge complexity, embrace uncertainty, and avoid falling back on simplistic cognitive models. Understanding the complexity of the issue and how people make decisions is key to having a fighting chance of solving the climate crisis.

**Keywords:** environmental change; decision-making; environmental awareness; environmental concern; action gap; mental model; theory of mind





## 1. Introduction

Climate, poverty, and biodiversity loss are global geopolitical affairs. The global economy, measured in GDP terms, reached 120 trillion USD in 2019 [1]. Global $CO_2$ emissions in 2019 were 3.3% higher than the year of the Paris Agreement [2]. Governments globally are planning to produce 120% more fossil fuels than the amount required to limit global temperature increase to 1.5 °C by 2030 [3]. The fossil fuel industry is planning to invest 1.4 trillion USD into extractive projects between 2020 and 2024. Meanwhile, tropical forest loss has accelerated compared to the baseline of 2002–2013 [4]. The anthropogenic pressures on biodiversity have sharply increased with one million species estimated to be

threatened with extinction [5]. Scientific evidence reports on alarming levels of Earth's climate system [6]. We are on a collision course with the earth system, a situation only slightly and temporarily slowed by the COVID-19 pandemic [7].

The magnitude of the change suggests that if anything, science was conservative. These signals are more severe than anticipated and point to our responsibility as sentient beings to consider the impacts of our actions. The most salient human drivers underlying these trends are increasing human and livestock populations, dietary habits, continued fossil fuel consumption, and global tree cover loss [8–12].

Unprecedented numbers of people have taken part in climate marches and protests across the globe in 2019. Amidst this civil outcry and mounting public pressure, the United Nations Framework Convention on Climate Change UNFCCC (CoP25), considered the last opportunity to establish clear rules on the implementation of the Paris Agreement before it came into effect in 2020, failed to deliver. The accumulation of failures due to ineffective actions has resulted in the loss of resources, time and reputation. It has fostered frustration, mistrust and contributed to the polarization of the debate. In that sense, ineffective action is potentially worse than inaction.

This urgency compels us to rethink decision-making in the context of a global environmental crisis. Drawing on concepts from distributed artificial intelligence [13] and behavioural and cognitive sciences [14–21], we articulate the factors that influence how individuals think and act with respect to the climate crisis through four hypotheses and propose a framework to make sense of how those factors interact. A key departure from most research on climate/environment/sustainability decision-making done so far is that we leave aside the empirical identification of drivers behind decision-making. In doing so, we also abandon to the decision-makers the full responsibility of their choices. We propose instead a theoretical model that helps understand decision-makers' behavior when competing information, worldviews, interests, and powers are involved. This framework serves to understand humans regardless of gender, culture, affluence, or credos. After all, we all make decisions relevant to the climate urgency [22–25]. The framework we propose allows anyone to make sense of what appears to many as irrational—the action gap (Figure 1). Finally, based on the logical implications the framework outlines, we propose interventions that can help to close the action gap.

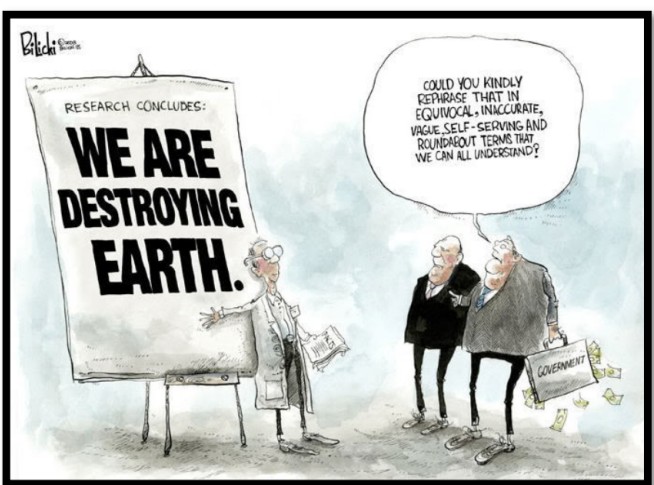

**Figure 1.** The action gap or the uncomfortable truth. The scientific community has been documenting the trends and ringing the alarm bell for more than 30 years. Yet governments and corporations seem not to hear the call. This successful, nonsensical drawing from 2007 was still relevant in 2020. Can we actually make sense of the situation it depicts? © Union of Concerned Scientists/Justin Bilicki, with permission.

## 2. Conceptualizing Our Choices

### 2.1. Multi-Agent System as a Model of Environmental Decision-Making

Derived from the field of distributed artificial intelligence, multi-agent systems are "systems that include multiple autonomous entities with either diverging information or diverging interests or both" [26]. Such a loose definition allows multi-agent systems to serve as a blueprint to represent societies of sentient agents interacting with each other and their environment. They provide a useful framework to conceptualize and discuss the critical processes that shape the agents' decisions.

Let us introduce at this stage Red, a fictional character in a multi-agent system (Figure 2). Red, as an embodied agent in the world, perceives her surroundings, space, resources, and other agents. She also shares information willingly or not with her surroundings. Red acts upon herself, the others, and what surrounds her. Her decisions and behaviour are shaped by her Umwelt as proposed by the philosopher Martin Heidegger (1889–1976)—the beliefs of the world that she trusts. Umwelt is normally translated as environment. More specifically, it is the "environment as perceived by the agent". We equate Umwelt with Mental Model—the set of beliefs about the world Red trusts. Choices made based on a flawed mental model might perform poorly, so in evolutionary terms there is considerable pressure to evolve the cognitive capacities that will help Red's mental model to be an accurate representation of the world [27]. However, human beings are highly constrained by their cognitive capacity and processing capabilities. As a result, our mental models are patchy, shallow, and often incoherent [28]. Mental models are efficient in that they are usually good enough to make effective decisions in a multitude of situations but perform systematically poorly in certain circumstances.

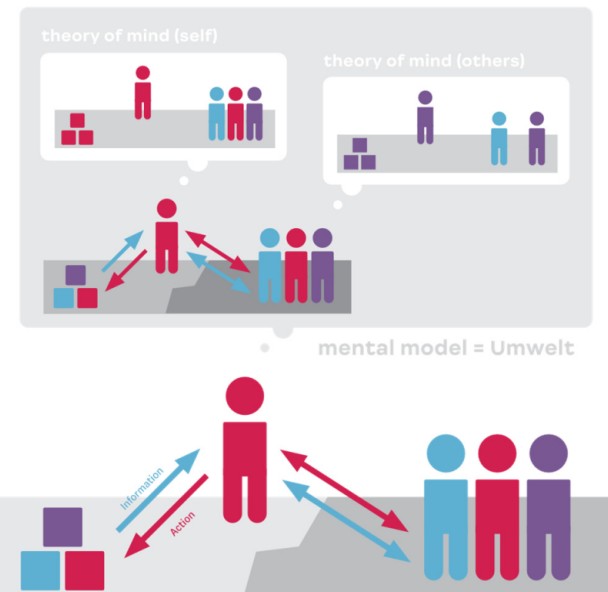

**Figure 2.** Mental models, Umwelt and theory of mind. This figure, normally used to describe a Multi-Agent System, serves as a blueprint to represent critical psychological processes that shape agency—our capacity to act independently and to make our own free choices—and decision-making. The red agent (Red) perceives her surroundings, the space, but also resources (blocks) and other agents, for whom she develops a theory of mind. Red exchanges information with these entities, can act with or upon them. However, her decisions and behaviour are based exclusively on her Umwelt. Figure inspired and redrawn from [13].

If Red attributes a state of mind—for example, intentions, beliefs, emotions, knowledge—to someone, Red has a theory of mind of that person. A theory of mind helps Red to understand that others may have different desires, beliefs, and perceptions of things from her own [29,30] (cf. Apperly [31] for a nuanced analysis on individual differences in

conceptual knowledge, cognitive processes and social competence influencing a theory of mind). It also makes it possible for Red to act in conjunction with other people. Given this definition, we propose to consider theory of mind as the equivalent of the mental model Red has of the mental model of an agent (Figure 2).

## 2.2. Reasons for the Action Gap

If climate change poses an existential threat, how do we make sense of our collective inaction—or worse, inefficiency, and how do we overcome this gap? Why do we do the things we do? Why do we fail?

The answer to these questions is not to be found in satellite imagery, companies' balance sheets or political programs. The answer is in the minds of those making decisions. As such, it is inaccessible to direct as well as indirect external enquiry, for a variety of hard reasons. These include (i) complacency (our tendency to answer what is expected from us), (ii) meta-agency (our capacity to answer what serves us best), (iii) the challenge posed by the distinction between contribution and attribution, and (iv) more importantly, the prospect of confabulation and the role self-attribution and post-hoc rationalization play in attitude formation and change—the fact that we find plausible explanations of our decisions not correlated with the drivers of said decisions [32]. We can nevertheless appeal to cognitive sciences to develop a comprehensive theory of mind of decision-makers—a model to explain and predict their behavior by attributing it to independent mental states, beliefs, and desires. Both facts and values play a role when making decisions and the latter is often missing from science communication [33]. We propose here such a model—a structure of ideas that explains and interprets the action gap. Four nested hypotheses are sufficient to account for the mismatch between the perceived urgency of the climate urgency and lack of efficient decision-making. Decision-makers do not respond efficiently to the climate urgency because they (1) do not hear the call, (2) reject its narrative, (3) do not share the concern, or (4) do not know how to respond.

## 2.3. The Archetype Framework

The four nested hypotheses form a full set of solutions—delineating all possible explanations for a decision-maker not to be effective. The hypotheses outline a bifurcating path for any agent to walk through, leading to five archetypes and strategies at their disposal (Figure 3). All archetypes operate with different information at hand, and under different values, beliefs, and capabilities. Using elementary concepts from set theory, it is possible to propose a mathematical formulation of these four hypotheses (Appendix A).

Let us follow Red as she moves through the path. The first bifurcation deals with the awareness of the decision-maker. Has Red heard of the issue? In the context of climate change, while technically possible, a 'no' answer is unlikely given the extensive media coverage and civil protests. It is difficult today for Red to be an Uninformed decision-maker (Figure 3) about climate change, biodiversity loss, deforestation, and a host of other environmental concerns.

The second and third questions relate to whether Red will dismiss the issue. A first question deals with her beliefs; does she accept the reality of the process, and the explanations provided by the scientific community? If Red answers 'no', she is a Denier decision-maker (Figure 3), and chances are she will fight or supress the narratives she denies. If Red answers 'yes', she must then confront her understanding of the world with her values: What matters to Red? There are many reasons why Red might consider the issue irrelevant or second to more pressing ones. If Red answers 'no' to the question "Does it matter?", she is an Occupied decision-maker (Figure 3). She must ensure her re-election, she needs to present the end of year figures to shareholders, or maybe she needs to ensure the well-being of her kids or her own survival—long-term thinking is a luxury not everyone can afford. Whether Red rejects the narrative or dismisses it for the time being, she will not share the concern of those advocating change.

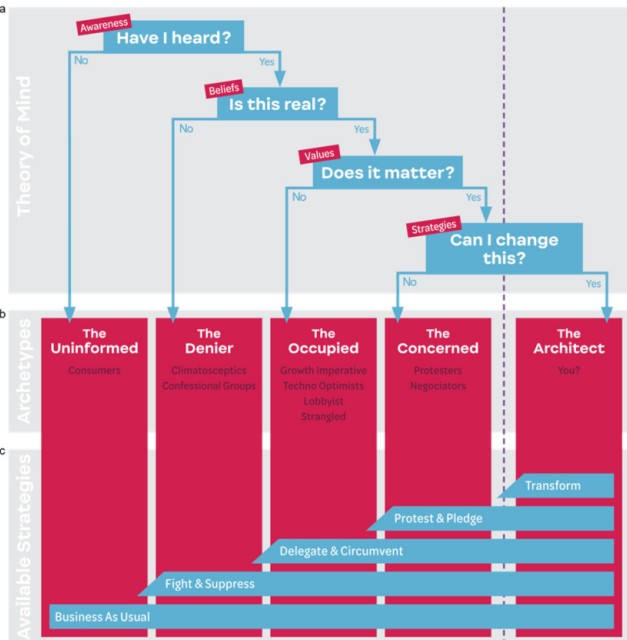

**Figure 3.** Choices we make: a framework to develop a theory of action (**a**). All archetypes (**b**) operate rationally under their conditions (e.g., social, educational); they strategize (**c**) with different information at their disposal, and under different values, beliefs, and capabilities. (Appendix B provides a non-exhaustive list of concepts that play a role in decision-making and the understanding of this framework).

The fourth hypothesis suggests Red does not know how to respond effectively. It asks the question of strategies and means. It forces us to consider that for these decision-makers, the root cause of the action gap might lie not in their willingness but in their inability to respond effectively. Their actions fall short or miss their targets. They do not allocate enough strength or fail to identify the correct levers for transformation. These are the Concerned decision-makers (Figure 3).

The shape of the deforestation curve [34] and the growth of $CO_2$ concentration [35,36] clearly indicate the final question in the series "Do I know how to respond?" has yet to be answered affirmatively at the global scale. Because of the pathologies of collective action we highlight—lack of awareness, lack of interest, lack of agreement, lack of common concern, lack of means—we see a world increasingly falling back on 'fend for yourself' national interests and geopolitics—an institutional form of the appeal to the privacy fallacy, or 'mind your own business'.

The framework has another crucial implication. It reveals the conceptual possibility of a fifth kind of decision-maker, a fifth archetype we call the Architect (Figure 3). It outlines the possibility of a decision-maker who is aware of the issue, accepts its reality and understands its causes, decides the topic is important enough to warrant action, and has found the proper strategy and means to actually redress the curve, at the global level. Do not misunderstand this for an appeal to a providential leader. The issue is one of collective action [37]. No single decision-maker has the power to redress the curve and the response must therefore lie in the development of effective alliances. Architects will enable the transition to happen, instead of forcing it.

## 3. The Consequences of Our Decisions

To understand the interplay of these different archetypes, we propose one more framework (Figure 4). Let us get back to Red—a decision-maker—and consider one variable of interest to her. Red's interest so far is unidimensional—her well-being. Red attaches a value to the fluctuations of that variable, giving a direction to that dimension, with positive and negative values. We posit the state of the world at present to be at zero.

Future developments might change the value of the variable, either for better (positive fluctuations) or for worse (negative fluctuations). The choices Red makes push the state of the world in the direction she prefers. We can represent this with a force vector, the direction of which is aligned with Red's intentions. The power of Red to shape the destiny of the world is represented by the length of Red's force vector. The more powerful the agent, the longer the vector (Figure 4).

We can conceptualize an ill-informed Red purposely pushing in the direction opposite to her interests. Such instance would entail that either there are other dimensions more important to Red—making the gambit the rational choice, even over the long-term and as long as she can sustain it—or that Red's misunderstanding of the world is such that Red's actions go directly against her main interests. This latter course will either be corrected—changing actions or changing interests—or Red will lose and disappear.

### 3.1. Red and the World

The world itself is not passive. Left to its own devices, it will move in directions difficult to foresee. The laws of entropy, the forces of physics at play, natural selection, self-organisation and all the possible biophysical processes push the world along the axis Red is interested in [38]. Some states of the world are more likely than others, some transitions easier than others. In other words, the world has its own vector. The playing field is not level and Red might have to go against the flow to increase the benefits she derives from the variable she is interested in. In other situations, Red might find it easier, as her interests align with the momentum of the world. Describing the internal drivers of the world is far from trivial and for the sake of clarity of the argument, we will only consider from now on the push of agents such as Red, and not the momentum of the world, or the slopes, pitfalls, and crests of the solution space. We will be content with linking these considerations to an informed discussion about the concept of resilience understood as the probability for a system to be in a given state [39,40].

Without awareness, there is no mental model of the problem and no solution space (cf. the Uninformed). If the causal links between human action and natural processes are in doubt, the solution space is unidimensional (the Denier). The other archetypes (cf. the Occupied, Concerned and Architect) perceive and recognize the interactions between humanity and the natural system (Figure 4a).

When the decision problem is multidimensional, all possible states of the world are represented as a hyperspace, the solution space, each dimension being one of the variables of interest. Red is gaining depth and complexity. We might want to consider for example all the Sustainable Development Goals (SDG) [42], each becoming a dimension to care for. It must be understood that the dimensions we decide to represent themselves contain multiple dimensions all encapsulated into the ones we decide to show. For example, a dimension of human well-being could be developed into as many dimensions as required to better describe the concept of well-being [43].

There are physical and natural constraints that we do not address here. The zero-sum game diagonal distinguishes destructive (Figure 4, bottom left) from constructive (Figure 4, upper right) strategies. Entropy and the second law of thermodynamics suggest destructive strategies are easier to implement than constructive ones. For the sake of simplicity, let us consider two dimensions, SDG 1 (Poverty) and SDG 15 (Life on Land). These goals are often perceived or represented to be in conflict [44,45], but the solution space gives room for all possible situations—trade-offs, win-win, lose-lose, and decoupling. We propose to represent SDG15 as the horizontal axis (Figure 4), with positive values representing more diverse, rich, and abundant land ecosystems, able to store more carbon. Negative values represent reduced carbon sinks, and the loss of biomass, of diversity, and in general of natural capital. Similarly, we will represent SDG1 as the vertical axis, with positive values representing wealthier, healthier, more secure livelihoods, and negative values representing increase in poverty, vulnerability, and the loss of human capital. This defines four quadrants Q1 to Q4 (Figure 4). Q1 represents outcomes where humans thrive at

the expense of the rest of nature. It could be argued that the story of human civilization since the end of the last ice age has been happening in this quadrant. Q2 represents outcomes where humans thrive in a vibrant ecosystem. Q3 represents outcomes where the human condition is worsening but the rest of nature thrives. Forest recovery in Chernobyl after humans left would sit in this quadrant. Finally, Q4 represents worsening conditions for all parties, ecological and social collapse hand in hand; an example here would be Easter Island [46,47]. Decoupled trajectories sit on the axis—true zero impact development running vertically, socially neutral ecosystem restoration landscapes running horizontally.

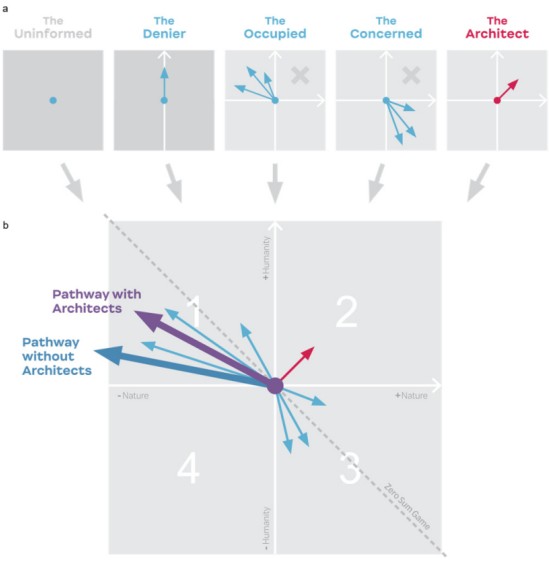

**Figure 4.** The solution space (**a**) is another framework to represent the development pathways each archetype is ready and willing to explore. There are four sub-spaces [41] for exploration: (1) humans win at the expense of nature; (2) win-win pathways; (3) nature wins at the expense of humanity and (4) all loose. Vectors are used to represent agents and their decisions: (i) the length of the vector represents the power of an agent; (ii) the direction reflects the intentions of an agent; (iii) the angle (compared to an axis) denotes the efficiency an agent establishes for the trade-off between the two dimensions (axes). The values an agent holds define the quadrants one is prepared to explore and the direction the vector takes. An archetype is averse to explore dark blue quadrants, favorable to light blue quadrants and does not consider crossed quadrants as possible. Occupied ones are willing to destroy natural capital as long as social capital increases. Concerned ones are willing to reduce social capital provided natural capital increases. (**b**) When the world is inert, the pathway the system will follow results from the sum of all individual vectors. Deniers and Occupied on one side and Concerned on the other are pushing in opposite directions. This represents the tug of war we want to break away from. The Architect can help move the system towards new directions. The playing field, however, is not level.

At this stage, we must acknowledge that the orientation of the axis is arbitrary. We choose an orientation that provides meaning. Different observers can choose an orientation that makes most sense to them by swapping the axes. The symmetries will conserve all the properties of the model. We are effectively proposing a value-neutral framework in the sense that it accommodates all possible subjectivities, including ours, and makes them visible, transparent, and therefore subject to scrutiny and debate. One more point, we consider humans as part of nature [48,49], and find it useful here to discriminate what happens to us from what happens to everything else in order to be able to conceptualize our relationship with everything else (Figure 4).

With all these elements in place, let us now consider how Red moves in the solution space as she walks down the path of the framework we propose (Figure 3; Appendix C for hypothetical real-world examples).

### 3.1.1. Uninformed Red

An Uninformed Red does not consider the variable of interest at all. She is unaware and uninformed. Her actions will register on the dimensions of interest, but it will not be the result of a positive choice. From her standpoint, the diagram does not exist. Only an external observer would be able to measure on her behalf the impact of her choices, or lack thereof, on the dimensions of interest. To all effects, lacking awareness of the issues, Red can be assumed to be part of the momentum of the world.

### 3.1.2. Denying Red

A Denying Red is aware of the issues and of the dimensions of interest. She purposely acts to increase her well-being, the dimension she cares for and denies that the other dimension is part of the equation. If Red is right, her actions will have no bearing on the dimension she rejects. If Red is wrong, her actions will have an impact but only an external observer would be able to measure it, most likely without Red's consent since she opposes the narrative.

### 3.1.3. Occupied Red

Moving up the next fork requires Red to change her beliefs (Figure 3). She now accepts the narrative of the other dimensions of interest and the existence of a causal relationship between the two. How and when this happens is the object of a lot of scrutiny by academics, think tanks, governments, and corporations [23,50]. We shall not elaborate on this question. Let us simply consider its implications. From now on, we would be able to talk of a multidimensional solution space with Red, and she would accept that—if she had the time or the inclination to ponder the issue, both things Occupied Red lacks. We will no longer wonder why scientists struggled for so long to grasp the attention of policy-makers on these issues. Occupied Red will strive to foster her main interest: satisfying her well-being. Red will be keen to explore Q1/2 and would be averse to developments taking the system towards Q3/4. If Red knew of a way to have her cake and eat it too, she would choose that option and explore preferentially Q2 (Figure 4). Very few of us consciously decide to destroy nature, eradicate species, and wreak havoc with the climate for the sake of it. We posit environmental destruction as overwhelmingly the indirect results of the choices we make. We can be oblivious to the consequences of our choices, deny them, deem them irrelevant or lament them, but they seldom are our main objective. Red simply does not know how to achieve both—and probably thinks it is not possible [51]. Red is Occupied, so will not spend time, resources, and energy to find solutions in Q2. Occupied Red, operating rationally within the boundaries of her mental model, will therefore explore Q1.

At this stage, we can introduce one more element. The angle of Red's vector (Figure 4) denotes the efficiency of the relationship Red establishes between the two dimensions. A vector close to the horizontal (x-axis) translates a wasteful use of natural capital for a marginal gain of well-being. We call these destructive (catabolic) strategies. A vector close to the vertical (y-axis) is an efficient use, allowing a significant gain in well-being at a small environmental cost. We call these constructive (anabolic) strategies. Occupied Red has therefore an infinity of strategies to choose from, some constructive and some destructive, but they all point towards Q1. All the choices she will make shall contribute to worsening the environmental situation. Occupied Red cannot bend the curves of environmental degradation; she can at best slow them down.

### 3.1.4. Concerned Red

Eventually, as the environmental crisis unfolds, Red might finally decide that change must happen, and must happen now. Concerned Red is now willing to shift her main object of interest from her well-being to the state of nature. This is a momentous change. Red is willing to endure personal discomfort for something she deems larger than herself. Here again, the reasons that push Red to shift priorities is not our concern. We are content to consider this change can and does happen and leave others to explore the causal links

behind such a personal transformation. Concerned Red is willing to explore Q2/3 and will be averse to Q1/4 (Figure 4). Given the choice, Concerned Red might rather opt for Q2, but as with Occupied Red, she does not really believe this to be a credible option. Red will thus move towards Q3—degrowth, depopulate. She can defend this position because fighting inequity and poverty and providing better quality of life are positive developments. Red might also consider that the future well-being of humanity depends on making the state of nature the main criteria now—the pathway to Q2 going through Q3 until a balance is found again. As with the Occupied, the angle of Red's vector denotes the efficiency of her actions. Vectors close to the vertical create a lot of strife and discomfort for a marginal environmental gain. Vectors close to the horizontal greatly improve the state of nature at a fraction of the cost for well-being.

### 3.1.5. Red and the Others

We can now move to the final stage of the framework (Figure 4). Red is not alone in the woods. She is part of a society of agents, each with their own force-vector representing their interests, power, and choices. The society is thus formed of an unknown proportion of Uninformed, Denying, Occupied and Concerned agents (Figure 5). The sum of all their individual vectors will condition the future state of the world (Figure 4). Remember we assumed for the sake of simplicity the world to be motionless and moving on a level playing field. Without a critical mass of Concerned agents, there is simply no alternative to Q1. The trends of environmental degradation are locked-in, and the best we can hope for is a slowing down as efficiency gains are made here and there, progressively aligning the vector with the y-axis. As the planetary boundaries are crossed and as the ever-earlier planetary footprint day [52,53] suggests, the ecological basis on which humanity stands might not endure for long [54]. With environmental degradation, vertical vectors denoting constructive strategies will become increasingly difficult to maintain [55]. At the limit condition, once we cut the last tree, we will only be able to warm ourselves by that fire for a little while—a dire loss for nature, a meagre comfort for humanity—before plunging into darkness. Unless the scientific community is wrong, we might have to consider Q4 as the new normal.

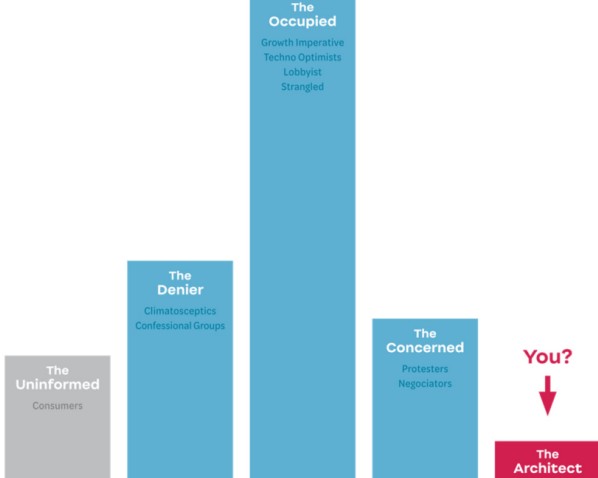

**Figure 5.** Society is made of an undetermined proportion of Uninformed, Denying, Occupied, Concerned citizens and (maybe) a few Architects. These proportions fluctuate with time and the accumulation of first-hand evidence, counterfactual narratives, mounting costs and the rise of concerns. How people transition from one archetype to the other becomes a crucial point of contention for anyone intent on bending the curves of environmental degradation. Vested interests and activism work to prevent such transitions or promote them actively. The first Architects can serve as catalysts for change.

*3.2. Lose-Lose Scenario as the New Normal*

A critical mass of Concerned citizens with sufficient power can take us in a different direction. Depending on the strategies they are able to deploy (Figure 3c), we might move right or down in Q1. With sufficient pulling power, they might even carry the world towards Q2 and Q4 (Figure 4). However, one of these outcomes is more likely than the other. As the Occupied will find it easy to develop inefficient and destructive strategies turning a lot of natural capital into a marginal benefit for humanity, the Concerned will find it easier to create strife for marginal ecological benefits. It takes a very imaginative activist to save the planet without pain. This is a particularly deleterious combination. A society of unimaginative but determined Occupied and Concerned agents, all developing destructive strategies take us straight into Q4.

For now, when it comes to the climate urgency and the biodiversity crisis, it seems that the combined pressure of all the Uninformed, Denying and Occupied agents outweighs the weight of the Concerned decision-makers (Figure 4). We are in Q1, maybe on the verge of moving towards Q4 [56,57]. It takes changing the minds of millions of people to push the world into Q3.

Is Q2 a possibility? Not if the world is a zero-sum game (sensu Weinbull [58]). In such a case, the possible space of solutions is bounded by the bisecting diagonal (Figure 4b), and gains in one dimension must be compensated by losses in the other dimension. More to the point, if people (e.g., Occupied, Concerned) believe the world to be a zero-sum game [51,59], they are unlikely to explore solutions in this direction. These solutions do not exist yet, or else we would have adopted them. They need to be invented, developed, constructed. We might stumble upon them unwittingly, but cards are stacked against that. Believing in their possibility is, however, a first step in the right direction.

The final stage up the path we have outlined has Red finding ways to reverse the trend. Red then becomes an Architect developing strategies that allow us to explore Q2. If the world is not a zero-sum game for the variables we chose to explore, then Red as an Architect is a logical possibility, and we must do everything we can to foster her emergence. As with the others, the angle of Red's vector will indicate whether she is focusing on well-being or on the environment. Red's power will not counterbalance but shift Occupied and Concerned towards more constructive pathways, breaking the tug of war between these two groups (Figure 4b). More importantly, having found such a way, all the others (e.g., Concerned, Occupied) having strategies (Figure 3c) less or equally efficient to hers will have a better alternative at their disposal—it is easier to copy than to invent. We would witness a shift of the individual vectors aligning with Red's, the transition happening when the sum of all vectors moves from Q1 to Q2. Red as an Architect is the alternative to the wars and collapse of Q3 and Q4.

## 4. The Way Forward

We can now answer the question: Why haven't we reversed the trend until now, despite the scientific evidence and the clamour from the street? Because we collectively did not know, did not believe, did not mind, or did not have the power to do so. Because we lack Architects with enough power among us. How do we continue? If we want to change things, we must give power to future Architects or become Architects ourselves.

How do we become Architects then? As we said earlier, climate change, deforestation, biodiversity loss, and related environmental issues are wicked problems. People typically think they understand complex problems far better than they do [60,61]. Take the example of tree planting. Tree planting is now advanced by many as a technical fix to the environmental crisis. Planting trees has effectively become a cognitive shortcut to a systemic problem, a kind of displacement behaviour if adopted by the Concerned, at worst a red herring used by the Occupied and the Denier. Architects would enable planting only when it makes sense, for example. The first step is therefore to shatter our illusion of understanding [61]. This can be achieved by developing plausible, mechanistic narratives of future developments. Such an exercise measurably reduces polarization [61].

The second step is to enrich our mental models about us, the others, and the world [62]. We do that by bringing in, and considering in our mental model, the elements that generate complexity such as causal relationships, agency, bounded rationality, and multiple feedback loops. In particular, this means refining our mental model of the self and of the others, shedding simplistic or intuitive theories of mind and adopting a better representation of human agency [26]. With a more nuanced, complex and rich understanding of how and why we think and act as we do—requiring caring and cognition of oneself and the others—it becomes possible to shift perspectives, to better understand divergent points of views [63]. We argue that having the present framework and the archetypes in mind is a step toward becoming an Architect. A better representation of the physical and natural processes at play (e.g., climate–forest–food–health nexus), of the coping strategies and agency of the stakeholders involved, and of their own capabilities and interests, will help decision-makers develop more pragmatic resilient and inclusive strategies through explicit theories of change [34,64].

The framework suggests only Concerned decision-makers can become Architects. As long as decision-makers depend on their political base, it is important to shift up the path (Figure 3) not only leaders, but followers as well. For leaders to become Architects requires power, and one way to empower Architects is through elections [65]. Climate-related risk now rank first on the 2020 Global Risks Report [66], suggesting a critical mass of Concerned citizens is gathering. As we have said, 2019 marked a turning point in the public awareness of climate change, and future events—new fires, unbearable temperatures, melting permafrost and deadly floods—will strengthen that realization. Past events, however, show that the protests can lead to lose-lose scenarios. Unless we fundamentally reform the process of decision-making, we risk remaining trapped below a zero-sum game between humanity and nature, between growth and equity, between their interests and ours (Figure 4). In the face of urgency, targets matter less now than actions. Intentions less than efficiency. Objectives less than pathways.

## 5. Conclusions

The pathway Red walks in is primarily an appeal to introspection. Ulterior motives or intent are difficult to determine, and it is challenging or even impossible to unambiguously assign a person to any single archetype. There is no way to tease out greenwashing from inefficient action. It is even likely that multiple archetypes co-exist within a single person. We are large, we contain multitudes. It is, however, useful to have the full theory in mind and realize that all archetypes conceptually exist. One of the more vexing elements of the climate action gap is its apparent irrationality. For anyone involved in the discussion, no matter where they might stand, the choices and attitudes of others seem disconnected from reality. Yet when we deem someone irrational, we suggest we cannot understand her choices. Here, outlining this framework, we make sense of what appears irrational. We explain the action gap. We must realize the reason for the action gap might be one of wrong strategy or limited capacity, and not just one of conflicting values or beliefs.

We are aware of two limitations of this framework. The first one is that we all navigate between the archetypes. Red shifts across them based on the questions she is asked, the decisions she makes and the conditions that prevail when her decision is made. We say little here of the reasons why Red would shift from one to the other. How do we become aware of a new topic? How do we come to accept new facts? How do we change values? How do we invent new strategies? We cannot do these questions justice here and they are left for future development. The second one is the meta-agency of people, who can choose as a strategy to present to the world an attitude that is not the one they adhere to. An Occupied candidate can pretend to be a Denier to secure votes. A Denying activist can masquerade as a Concerned to fuel discontent. A Concerned employee can pretend to be Occupied with the welfare of the company to climb up the ranks unopposed and change things from within. This Machiavellian approach to the public discourse does not invalidate the framework but invites caution when attributing intent to a public figure.

We have consciously left space in our narrative for doubt—the framework we propose remains valid even if climate change is not an existential threat. What matters is whether we consider that it does. Understanding each other is a step towards more constructive dialogues. The proposed framework goes in that direction. Climate, poverty, and health are global affairs, which do not stop at political boundaries nor follow ideological trenches. A possible next step involves making the mental models behind decisions transparent and subject to public debate. Generating and validating mechanistic explanations of the impacts—intended or not—of contemplated policies, eliciting the assumption and mental models decision-makers operate under—a radical new form of transparency—would lead to more constructive democratic discussions [61].

To redress the curves of environmental destruction, we need to change the way we make decisions. To do so, we first need to understand how we and others make decisions. This framework allows to better understand behaviours in the context of the environmental crisis we are facing. It is a first step toward becoming an Architect.

**Author Contributions:** Conceptualization, P.O.W. and C.A.G.; writing—original draft preparation, P.O.W., C.A.G., N.S., L.W. and J.G.; writing—review and editing, N.S., J.D.L., J.S., C.N., J.L.I., P.F. and S.A.S.; validation; P.F., J.S. and S.A.S.; visualization; P.O.W., C.A.G., L.W.; supervision, P.O.W. and C.A.G. All authors have read and agreed to the published version of the manuscript.

**Funding:** This research received no external funding.

**Institutional Review Board Statement:** Not applicable.

**Informed Consent Statement:** Not applicable.

**Data Availability Statement:** Data sharing is not applicable to this article.

**Acknowledgments:** Figure 1 by Union of Concerned Scientists/Justin Bilicki (2007) with permission. All other figures designed by Sylvain Mazas (sylvainmazas.net/info). We are grateful to Red for the ride.

**Conflicts of Interest:** The authors declare no conflict of interest.

## Appendix A Set Theory

This framework, and the nested hypotheses that structure it, allow for a mathematical formulation based on elementary set theory. Let X be the set of decision-makers, and x be an element of X. Let us ask a first question "Is the decision made by x effective?" We will call A the subset of X of effective decision-makers, and $A^c$ its complement in X, the subset of ineffective decision-makers. The action gap suggests the size of A to be much less than $A^c$

$$|A| \ll |A^c| \tag{A1}$$

Or possibly even,

$$A = \varnothing \tag{A2}$$

Within $A^c$, we seek to answer the question: "Why is x not taking effective action?". The 4 hypotheses above outline 8 nested subsets, U and its complement $U^c$ in $A^c$, D and its complement $D^c$ in $U^c$, O and $O^c$ its complement in $D^c$, and C and $C^c$ its complement in $O^c$.

Each subset is defined by a property of as follows:

U is the subset of decision makers that have not heard of the issue and $U^c$ the subset of informed ones. Within $U^c$, D is the subset of decision-makers denying the reality of the information received and $D^c$ the subset of accepting ones. Within $D^c$, O is the subset of decision-makers unwilling to address the issue and $O^c$ the subset of willing ones. Within $O^c$, C is the subset of decision-makers unable to take effective action.

We have

$$O^c = C + C^c \tag{A3}$$

$$D^c = O + O^c \tag{A4}$$

$$U^c = D + D^c \tag{A5}$$

$$A^c = U + U^c \tag{A6}$$

We propose

$$C^c = \varnothing \tag{A7}$$

An informed, accepting, willing and able yet inefficient decision maker is not a logical proposition. If they are inefficient, they are not able. If they are able and willing, they are efficient. The case in which the actions of others counter the decision of such a decision-maker, will be covered later—suggesting the possibility of invisible Architects.

It follows that

$$A^c = U + D + O + C \tag{A8}$$

And that

$$X = U + D + O + C + A \tag{A9}$$

While Equation (A2) is true, we know why we are unable to effectively address climate change. The four sets U, D, O and C form a partition of $A^c$, such that every decision-maker x is in exactly one of these mutually exclusive subsets. The model covers all possible cases and sufficiently and unambiguously describes the set of inefficient decision-makers. We have failed so far because we are collectively uninformed, denying, unwilling or unable to act.

To effectively address the climate urgency, Equation (A2) must be wrong. We need effective Architects. U, D, O, C and A will then in turn form a partition of X. This is, however, a necessary but not sufficient condition. There is a critical mass of Architects required for their combined power to push the system toward preferred futures.

**Appendix B How We Make Decisions—A Primer**

The following section serves to illustrate that the way that we make decisions can explain how the action gap came about. It is not intended to be representative of the most important theories of decision-making, nor is it comprehensive.

In the Anthropocene, Earth system transformations are the consequence of choices, outcomes, and strategies. Choices refer to the ability an agent has to decide on an action from a range of possible options. Outcomes are the result of the interactions between the choices made by the agent and the rest of the world. Strategies are the sequences of choices the agent makes (Figure 3a,c).

On what grounds do we make choices? Decision-makers are constrained by their awareness of the issues, beliefs about how climate change works and how it pertains to their environment, personal values, and by the political, social, economic and technical conditions under which they are operating [67–71] (Figure 3a). Beliefs are central to our understanding of choices [33]. They are formed to inform judgment, reasoning, communication, decision-making, but also because we value them for their own sake [72]. In psychology, decision-making refers to the cognitive process involved in choosing between a set of possible alternatives. This choice is made based on the values, preferences, beliefs, knowledge, and emotions of the decision-maker when going through this process [73,74]. Beliefs are updated on a continuous basis through external and internal cues [33,72]. Surprise is a crucial element of this monitoring process. The function of surprise seems to be the detection of unusual events that require either further processing—requiring attention—or fast reaction [75]. The accumulation of surprises can lead to epiphanies and sudden changes of beliefs and behaviours [76]. Humans are also a tribal species—beliefs are shared across our social network. This can lead to differences in beliefs between generations, communities, or political affiliations [22]. Beliefs are also communal construction, and much of our mental model [27]—the beliefs about the world that we trust—sits in the heads of the experts that we surround ourselves with and trust [60]. On issues related to climate change, attitudes are often strongly ingrained because the issues are complicated, hidden from view, and so slow to unfold that they are often imperceptible [77–79]. Thus, beliefs depend almost entirely on one's community [37,80–83].

Game theory, the Iterated Prisoner Dilemma (IDP) and the discovery of zero-determinant strategies that create the evolutionary conditions for the emergence of a theory of mind indicate that the way we come to a decision matters also [84]. When agents play the prisoner dilemma, the outcome of their decision is dependent on the payoff matrix and the decision of the other agent. When the problem is iterated, and agents are pitched against each other repeatedly, the way they come to decide on their next move dictates the success of their strategy—there is an evolutionary pressure to develop better strategies than incorporate the payoff matrix, the past behaviour of the opponent and its likely future choices—its values and interests [26]. In an IPD, the way a sentient agent comes to a decision matters more than the individual choices made along the way, even when each decision contributes to carve future path decencies. Certain pathways are more likely than others, but each new decision modifies these probabilities. The lesson we draw from the IPD is that in order to improve the quality of the strategies we develop, we have to expand the elements of the system we consider when making choices. Looking beyond our immediate interests to consider the interests of the others [84] and expanding the set of others we consider—from our immediate neighbours to the ones far removed, down to the voiceless and even the non-sentient components of the system [85].

Finally, even once we show willingness to accept an idea and share a common concern—climate urgency here—why can it happen that we are unable to respond? Decisions are never made in a vacuum [86]. We are part of a network of agents (Section 2.1); we share beliefs, desires, and aspirations, and we see others as allies, third parties or competitors. Climate change and other environmental problems are wicked problems (sensu Rittel and Webber [87]). There is no "easy" strategy to pursue, no straightforward decision to be made because wicked problems lack agreed upon definitions. They are complex and complicated (sensu Pietronero [88]), evolving and adaptive and have no stopping rule. Solutions to wicked problems are not right or wrong, they are instead more or less acceptable to the different stakeholders involved. Decisions to act or to wait have unknown consequences, potentially creating further challenges down the road. A way to deal with them is to "muddle through", observe, learn, and adapt [89,90]. Responses to wicked problems are social processes, and knowledge alone will not solve the issue. Unless we recognize the true nature of the problem, it is difficult to develop effective strategies.

**Appendix C  Meet Red—How to Use the Archetype Framework**

Here we provide a series of fictional examples that can help the reader better understand how the framework applies across sectors and hierarchies.

Example 1. Red as a decision-maker in an administration is constrained by the rules and regulations and the hierarchical structure in place. As a Denying civil servant, she would actively defend said rules even when they contribute to cause environmental damage, with even more determination as objections grow. As an Occupied civil servant, she could be content to follow procedures or accept them since they secure more important goals. As a Concerned citizen she might be unable to contribute constructively to the transition desired. Displacement behaviour would be her best answer to cope with the frustration and cognitive dissonance she is under. Unless she gains power or breaks the rules, she cannot explore a different quadrant. One of her superiors, however, could act as an Architect, empowering her to invent or develop efficient actions. Red could in turn enable others under her to act likewise, becoming herself an Architect. The responsibility is not equally shared here, since what we do is constrained by what those with power over us let us do.

Example 2. Red as a structural engineer who needs to decide where a dam will be located. As an Uniformed Red, she would take into account costs, transportation and the size of the reservoir. No environmental aspect would be considered in the decision. A Denying Red would probably reach the same decision but could also take steps to undermine the position of contradictors. An Occupied Red would accept the environmental damage as necessary evil. A Concerned Red would resist the company pressure and look

for ways to avoid environmental damage—up to a point. She would either end up quitting as a protest or give in and become Occupied. Unless she finds a way and as an Architect she manages to transform the mindset and modus operandi of the Company, generating profits while restoring nature.

Example 3. Red as a farmer in a deforestation front in the tropics. Uninformed Red minds her own fields and contemplates how to secure the future of her kids. She might start opening new fields next year. Denying Red does not listen to what scientists and NGOs are saying, and they better stay out of her land. Who are they to give lessons after what they have done in their country? Occupied Red has seen the changes. She remembers the insects flying in the fields of her youth. But that doesn't matter. She is now producing three crops a year and she is saving to send her kid to the University. She is not taking chances. Concerned Red also remembers how things were, but unlike her Occupied neighbour, she doesn't like what she sees happening around her. She is trying to do things differently, but is it difficult. The prices are low, customers don't pay for the biodiversity in the meadow and the interests of her loan are crushing her. She endures because she sees changes in her land. But for how long will she manage to continue? Red Architect managed to develop lasting relationships with customers that let her restore part of her estate, giving back some space for nature. The right subsidies made it possible to abandon parts of her activity she did not like for their environmental cost. If only her neighbours would join her.

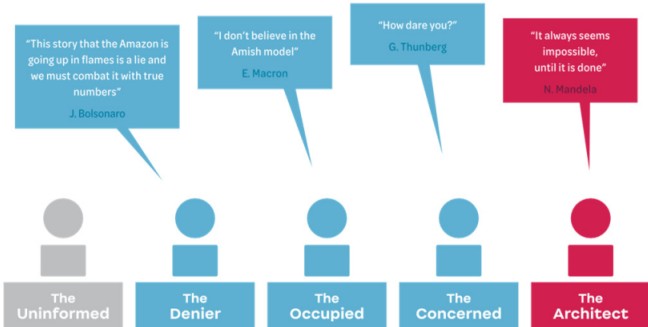

**Figure A1.** Quotes attributed to famous citizens. Keep in mind it is not possible to attribute unequivocally and irrevocably a person to a given archetype. One can only say at that moment a given person spoke as an archetype would speak. It matters because it helps identify if the argument is one of information, beliefs, values or means. The Uninformed, of course, are silent. NB: The quote of the Architect is attributed to N. Mandela but there is no evidence that this attribution is accurate. We nevertheless choose to retain it since it perfectly embodies the fight Architects need to put up against disbelief from all parts.

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
