# Peer review of "Choices We Make in Times of Crisis"

_sustainability, doi:10.3390/su13063578_

Round 1

Reviewer 1 Report

We appreciate the theme and approach proposed by the authors, which opens the way for reflections, towards new directions of action in the decision-making process.

Defining the classification of the paper in one of the typology announced by the journal (review or perspectives), would be appropriate for the identity of the text proposed by the authors.

We consider it necessary to be assigned a name for each figures used and to be moved the current explanations (from their actually presentation) in the text of the paper, as the current form of a figure's comment is sometimes too loaded (eg fig. 3).

We recommend the use of the specific citation of the journal, by using the parentheses for the indicated references, (eg [1,2]) giving up the current form of indicating the references by index.

Author Response

Dear Editor and Reviewers,

Thank you for your constructive feedback, which has helped us improve our piece. We have taken the time to carefully consider and respond to the comments made. Below are the detailed explanations of our revisions, based on your concerns and recommendations.

Reviewer 1

We appreciate the theme and approach proposed by the authors, which opens the way for reflections, towards new directions of action in the decision-making process.

  • Defining the classification of the paper in one of the typology announced by the journal (review or perspectives), would be appropriate for the identity of the text proposed by the authors.

Authors: We believe the best format for this paper would be a Perspective.

  • We consider it necessary to be assigned a name for each figures used and to be moved the current explanations (from their actually presentation) in the text of the paper, as the current form of a figure's comment is sometimes too loaded (eg fig. 3).

Authors: We have moved the unlabeled figure out of the former Box 1; it is now the new Figure 2. We have also shifted some caption text (new Fig. 4) into the body text.

  • We recommend the use of the specific citation of the journal, by using the parentheses for the indicated references, (eg [1,2]) giving up the current form of indicating the references by index.

Authors: done

Reviewer 2 Report

I have read the manuscript "Choices we make in times of crisis" with much interest. The manuscript covers a framework to describe decision-making in the context of climate change and develops an architecture of actions. The authors have made use of the literature from a variety of sources and different branches of the literature. It is also quite evident that the authors are accomplished writers and senior scholars. 

I had, however, the following doubt as I read the manuscript. The authors base the framework primarily on the notion of climate change and how a decision-maker may address this. However, the real-life challenge is that no single decision-maker is ever asked to singularly address 'climate change'. The authors would agree that climate change may manifest itself as a part of a complex problem, now a deforestation issue, now a failed crop, or a drought or flood. Therefore, a large part of the developed framework would not apply, given that a local level or provincial decision-maker would not be asked to address 'climate change' and a multilateral decision-maker would sit in a specialized agency where awareness and belief would not come into play. Therefore, in my view, no single decision-maker has the mandate to singularly or unilaterally address this and faces this dilemma. Issues faced by singular decision-makers are often incremental and not grand with immediate effect. 

The other part that the framework does not directly address is that of trade-offs. The choices made in the values are not necessarily between growth and development, but rather a more complex decision related to the width of a road or the location/ height of a dam. 

Perhaps the authors may consider identifying some limitations of the framework. 

Some minor points include: 

Lines 67-70: if the protests took place in 2019, by then it would be too late for the protests to have an impact on the outcomes of CoP25. 

Page 3 in general: I see the premise of the argument that the authors are setting up as one of the multilateral settings (the UN etc, as introduced in the previous section). However, this framework ignores that the decisions made in multilateral settings are not unilateral. Certain decisions are dependent upon the nature of the agency where the actor is placed. 

Page 3 in general: An important point: the amount of agency a single individual may have in decision making may be rather very limited. 

Finally, apologies if my review may have delayed the process. 

Author Response

Kind regards

Patrick Waeber

Reviewer 3 Report

Overall, while I don’t object to an application of something like a hierarchy of effects model to understanding what kinds of barriers exist to climate action, I found the presentation of the argument to be quite unclear.  While none of the 4 hypotheses stated for why people may not take climate action are unusual, there is little support provided in the manuscript for why each might occur.  While this may not be strictly necessary given the scope of the paper, I feel that it would improve comprehensibility and coherence of the argument

The manuscript lacks coherence in many places, sometimes appearing to contradict itself: for instance lines 80 to 81 reads “this framework serves to understand humans regardless of gender, culture, affluence or credos” while line 109 reads “Understanding people’s behaviors and cultural contexts can play a key role in shaping the response to the climate crisis.  We propose here such a model…”.  These points imply that the model both applies regardless of culture but also relies on cultural contexts which, while not necessarily impossible, requires a more thorough and coherent explanation than is currently present in the manuscript.  The coherence is not aided by the substantial number of grammatical errors in the manuscript.

Generally I find the literature review somewhat haphazard, particularly around section 2.2 “how we make decisions”.  While I suspect that the concepts surrounding Prisoner’s dilemmas, game theory, wicked problems, beliefs and norms are all valuable for understanding human decision-making, they are presented rather jumbled together.  It is not clear from this section why each concept is relevant to the framework and why these concepts were chosen over others (for instance, dual-process theories, heuristics and biases explanations, naturalistic decision-making theories etc.).  While there is clearly no need to provide a comprehensive review of the literature around decision-making, it would be useful to have a guide as to why these concepts were chosen and how they are relevant to the present effort. Given the framework that the authors are proposing, tying specific elements from the literature review to particular decision-making issues would strengthen this section considerably.

Finally, while there may be some merit in the description of the action gap using strategy games, it is not done clearly here.  Restructuring the paper around a more coherent narrative of how the proposed strategy game operates and how it correlates with policy and actual human action using examples might help aid in comprehension.  Making the perspective that the authors are taking using strategy games more obvious from the beginning of the manuscript may also help the reader follow the argument.

Minor issues:

Lines 71-72: The sentence beginning with “the urgency” should perhaps read “this urgency”?

Lines 75-79: this sentence is rather long, suggest breaking it up.

Line 95: it’s not entirely clear to me why collective inefficiency would be worse than collective inaction.  In the event that you want to propose that taking actions we think are effective but are inefficient is a problem for addressing climate urgency, it would be good to introduce that idea earlier on.

Line 121: should probably be: “…decide [on] an action”

Line 161: It may be a strong statement to say that magnitude of change is “proving” anything but this may simply be a stylistic decision.

Line 198: should probably be “with [a] mental model”

Author Response

Kind regards,

Patrick Waeber

Round 2

Reviewer 3 Report

Thank you to the authors for taking the time to address my comments.  I feel that the revisions made to the manuscript, particularly the relocation of the section on decision-making and adjustments to the presentation of the main arguments, have dramatically increased its coherence and clarity.